# Discovery of Personalized Treatment for Immuno-Metabolic Depression—Focus on 11beta Hydroxysteroid Dehydrogenase Type 2 (11betaHSD2) and Toll-like Receptor 4 (TLR4) Inhibition with Enoxolone

**DOI:** 10.3390/ph18101517

**Published:** 2025-10-10

**Authors:** Harald Murck

**Affiliations:** 1Department of Psychiatry and Psychotherapy, Philipps-University Marburg, Rudolf-Bultmann Straße 8, 35039 Marburg, Germany; murck@staff.uni-marburg.de or haraldmurck@yahoo.de; 2Murck-Neuroscience LLC, 525 South Chestnut Street, Westfield, NJ 07090, USA

**Keywords:** autonomic nervous system, vagus, blood pressure, nucleus of the solitary tract, aldosterone, 11beta hydroxysteroid-dehydrogenase type 2, toll like receptor 4 (TLR4), glycyrrhizin, therapy refractory depression, neuroinflammation

## Abstract

Treatment options for major depression are limited: only about one-third of patients achieve remission with first line treatments with no established predictive markers. Parameters associated with treatment refractory depression, including metabolic markers (increased BMI, increased triglyceride levels), inflammation markers (C-reactive protein, CRP), autonomic disturbances (reduced blood pressure, reduced heart rate variability), and brain morphology changes (increased volume of the choroid plexus and brain ventricle volumes), may serve such purpose. These features can be linked mechanistically to an increase in aldosterone plasma concentration due to a reduced mineralocorticoid receptor (MR) sensitivity. The primary CNS target of aldosterone is the nucleus of the solitary tract (NTS), which is also the entry point of the vagus nerve. This nucleus integrates signals from endocrine, inflammatory, chemoreceptive, and physiological parameters, including blood pressure. In search of a mechanism to overcome this pathology, we identified a molecule which is derived from the licorice plant glycyrrhiza glabra, namely glycyrrhizin and its biologically active metabolite enoxolone. These molecules potentially reverse the above-described pathology. They inhibit the enzyme 11beta hydroxysteroid-dehydrogenase type 2 (11betaHSD2) and the toll-like receptor 4 (TLR4). 11betaHSD2 regulates the activity of the mineralocorticoid receptor (MR) by degrading cortisol/corticosterone, which allows aldosterone to bind to the MR. TLR4 is the ligand for lipopolysaccharide (LPS, endotoxin) and trigger of innate immunity. Consequently, patients with increased inflammation markers, increased aldosterone, or low blood pressure may preferentially benefit from the treatment with glycyrrhizin/enoxolone. Importantly, these patients can be identified BEFORE treatment is initiated. Clinically, patients sharing these biological indicators are primarily young females or patients with a history of childhood trauma. A combination of enoxolone with standard antidepressants may therefore avoid a trial-and-error approach and allow to achieve recovery faster.

## 1. Introduction

STAR-D (Sequenced Treatment Alternatives to Relieve Depression), a large open label study in patients with depression revealed that first line treatment with the standard antidepressant, sertraline, a selective serotonin reuptake inhibitor (SSRI), leads to remission in only approx. 30% of patients [1]. This indicates that standard antidepressants only work for a small subgroup of patients. Established markers to identify these subjects do not exist, but a number of important demographic, clinical, and biomarker characteristics have been reported.

Neurobiological differences between clinical subtypes of depression with different treatment outcomes have been described. We focus on the distinction between melancholic and atypical depression. Atypical depression vs. other forms is associated with worse treatment outcome [2,3]. Atypical depression shows hypersomnia, hyperphagia, emotional irritability, and somatic complaints [4]. Melancholic depression shows opposite features, including weight and appetite loss, early morning awakening, psychomotor changes, and a specific melancholic mood, sometimes referred to as “feeling of feelinglessness”. Differential biological pathways underlie these subtypes, which involve endocrine regulation (hypothalamus–pituitary–adrenocortical (HPA) axis and possibly the renin–angiotensin–aldosterone system [RAAS]), metabolic and autonomic regulation, including control of blood pressure and inflammation [4,5,6,7]. Atypical depression has been recognized as a form, which goes along with a reduced noradrenergic activity and normal to reduced cortisol plasma concentration [8,9], whereas an increase in HPA axis activity and noradrenergic activity is related to melancholia.

Even though melancholic depression is the textbook form of this disorder, it appears to contrast with the foundational neurobiological explanation of the neurobiology of depression, i.e., the idea of a monoamine deficit in depression. This hypothesis was based on the efficacy of norepinephrine reuptake inhibitors and monoamine oxidase inhibitors [8], and the observation that reserpine, a compound which depletes noradrenaline, induces depression [10]. Interestingly, reserpine was utilized as a medication to treat hypertension, i.e., to lower blood pressure. One important adverse event of the compound is fatigue. An association between low noradrenaline levels, hypotension, and fatigue may indicate that the reserpine model reflects characteristics of atypical depression more than that of melancholic depression.

Several newer studies revealed a number of additional differentiating characteristics between atypical and melancholic depression, namely an increase in inflammation and metabolic characteristics, including an increase in body mass index (BMI) and triglyceride levels in the atypical form [11,12,13]. Interestingly, autonomic markers, including higher heart rate and low cardio-autonomic balance (derived from respiratory sinus arrhythmia and pre-ejection period; higher values are associated with higher parasympathetic activity), were predictive of lower high density lipoprotein (HDL) two years later, indicating that metabolic changes are consequences of autonomic vulnerability factors [14].

On the basis of these observations, a framework of immuno-metabolic depression was developed, which shows an overlap with atypical depression symptoms. However, the context of these biomarkers may be of importance; for example, elderly men, who are often melancholic, show an increase in inflammation, which may be related to a different mechanism than in young women [15], who preferentially show signs of atypical depression [16]. We will focus on the relevant biomarkers in the following, starting with a simple biomarker, blood pressure.

## 2. Blood Pressure

Patients with major depression tend to have a lower than normal blood pressure [17,18]. The study of Licht et al. [17] also demonstrated that standard antidepressants tend to increase blood pressure. Similarly, anxiety and depression were associated with low blood pressure in several large studies [19,20], including in elderly subjects [21]. An additional feature of low blood pressure is fatigue: this has parallels to chronic fatigue syndrome (earlier referred to as neurasthenia) [22,23].Vice versa, higher blood pressure is linked with lesser perceived stress and higher quality of life [24,25,26]. On the basis of the observation that higher blood pressure is associated with wellbeing, Dworkin formulated the hypothesis of “learned hypertension”: blood pressure can be upregulated based on the sense of an improved wellbeing via classical conditioning [27]. This hypothesis was largely confirmed on the basis of long term data [28].

Accordingly, low blood pressure is associated with an increased risk to develop depression [29] and, in particular, atypical depression. It may, however, be protective against melancholic depression [30]. Low blood pressure is also a risk factor for therapy resistance in depression [31], primarily in women [32]. This is also supported by data comparing outpatients with hospitalized patients with depression, the latter plausibly with more refractory forms, as these patients had a reduced blood pressure in comparison to outpatients [33]. These subjects showed high triglycerides and low HDL, which may have contributed to the assumed therapy refractoriness, as will be discussed later.

Dynamic changes in blood pressure in the course of the day should be considered. A drop of more than 10% of systolic blood pressure during sleep defines a so-called dipper status, which is associated with lower cardiovascular risk in comparison to non-dippers. Non-dipper status is often observed in patients with hypertension, who also show higher anxiety and depression levels [34], sleep disturbance [35], or shorter sleep duration [36], indicating melancholic features. However, one study found a more pronounced blood pressure dip [37] in patients with higher depression severity. Use of sedating medications in the study population was made responsible for that.

Support for a causal relationship between low blood pressure and depressed mood is implied by observations that pharmacological interventions with the alpha adrenergic compound midodrine [38] increased salt intake, or fludrocortisone has beneficial clinical effects. Fludrocortisone is an agonist at the receptor for aldosterone, the mineralocorticoid receptor (MR), and can improve orthostatic reactions [39] and depressive symptoms [40,41]. In addition, an influence of vagal function can be suggested from findings in patients with vasovagal syncope, who did not respond to pharmacological or non-pharmacological treatment [40]. These patients showed higher depression and anxiety ratings; even subsyndromal orthostatic hypotension is associated with cognitive dysfunction, increased hopelessness [42], and increased risk of anxiety and depression [43].

## 3. Renin–Angiotensin–Aldosterone System

Aldosterone is a steroid hormone best known for its blood-pressure-regulating effect. It is regulated via the sympathetic nervous system by beta adrenergic activation of the renin–angiotensin II cascade; in addition, in the short term, the HPA axis is involved, as ACTH also leads to aldosterone release (see [44]). Potassium can directly release aldosterone from the adrenal cortex. Mineralocorticoids, including aldosterone, have been recognized early to affect brain function, focusing on autonomic, but an effect on emotional regulation was not considered [45]. This is despite the fact that Selye, who brought the “stress” concept into neurobiology, pointed out the importance of mineralocorticoids, like aldosterone as proinflammatory and stress hormone [46]. Aldosterone’s role in emotional regulation became a topic of interest more recently [31,44,47,48,49]. The proinflammatory property of aldosterone has also been widely confirmed and may be part of its behavioral effect [50].

The role of aldosterone as a mediator to induce depression is well established in animal experiments: subchronic infusion of aldosterone leads to depression- and anxiety-like behavior in rats [47]. Furthermore, under acute stress and in experimental models of depression, including tryptophan depletion and magnesium depletion, an increase in plasma aldosterone occurred, which was associated with an increase in depression-like behavior [51,52,53].

In humans, acute stress leads to an increase in aldosterone in healthy volunteers [54,55]. By recognizing depression, at least partially, as a stress-related disorder, an association with major depression appears plausible. The fact of an increased level of aldosterone in patients with depression is now well established [56,57,58,59,60]. A causal relationship is implied by the observation that aldosterone leads to depressive and anxiety symptoms, as demonstrated in patients with primary hyperaldosteronism. Symptoms improve after its treatment, but with a fairly slow time course, indicating that long-term changes were initiated in the brain [61,62]. Similarly, polymorphisms of the renin–angiotensin–aldosterone system, i.e., of the angiotensin-converting enzyme (ACE) and the angiotensin II (ATII) receptor are associated with worse treatment outcome for the more biologically active genotypes [63], which are expected to be associated with higher aldosterone levels. Important in our context, a high ratio of saliva aldosterone/cortisol appears to be related to a less favorable therapy response [31] in hospitalized patients with depression.

Of note, the increase in aldosterone as observed with an aldosterone-releasing adenoma or aldosterone administration is associated with an increase in blood pressure and increased BMI. Increased BMI and increased aldosterone/renin ratio were positively, but independently, associated with the severity of depression symptoms [64]. Interestingly, in the latter study, an increase in blood pressure was associated with less anxiety in men, which may point to a counterregulatory mechanism of blood pressure on anxiety, possibly via baroreceptor activation. This is also in line with the above-described protective effect of a higher blood pressure against the development of depression.

What is the mechanism of aldosterone to induce depression? Aldosterone has a fairly specific neuronal target, i.e., nucleus of the solitary tract (NTS). The NTS regulates autonomic function. It projects to anatomical targets related to motivation, like the nucleus accumbens, interoception, like the insula, as well as prefrontal cortical areas, like the anterior cingulate [65,66,67]. Aldosterone acts at neurons, which co-express MR, and the enzyme 11-beta-hydroxysteroid-dehydrogenase type 2 (11betaHSD2). This enzyme rapidly degrades cortisol intracellularly and therefore lets aldosterone compete at the MR, which otherwise would be occupied with the much higher concentrated cortisol. In the absence of 11betaHSD2, as for example in the hippocampus, cortisol/corticosterone is the main ligand at the MR. These neurons of the NTS, which co-express MR and 11betaHSD2, overlap with those of vagus nerve afferents [68] and are involved in salt appetite [69]. This is relevant as salt appetite is increased in depression and anxiety [70,71,72] and can be used as a behavioral marker for central MR activation. The NTS is also activated via the baroreceptor, which indirectly inhibits sympathetic neurons in the rostroventerolateral medulla (RVLM) [73]. At the same time baroreceptor activation increases parasympathetic output. Aldosterone inhibits this mechanism [74], therefore activating the sympathetic and inhibiting the parasympathetic nervous system. Interestingly, baroreceptor activation may induce an anti-inflammatory action [75], which will be covered later. Another nucleus which may be involved in sympathetic regulation of aldosterone is the paraventricular nucleus of the hypothalamus, which, similar to the NTS, co-expresses 11betaHSD2 and MR. Inhibition of 11betaHSD2 leads to an activation of the PVN, which goes along with an increase in HPA axis activity, an increase in blood pressure, as well as an increase in renal sympathetic activity [76].

### What Leads to an Increase in Aldosterone?

Aldosterone release is activated by low blood pressure (via RAAS activation) and low plasma sodium. Low blood pressure as a trigger needs to be explored. A potential explanation for the lower blood pressure is the lower sensitivity of peripheral mineralocorticoid receptor (MR). MR are involved in vascular compliance and the absorption of water and electrolytes, both of which affect blood pressure. We observed evidence for the desensitization [31] as the correlation curve between aldosterone concentration and blood pressure was shifted to the right in non-responders to standard treatment. This means that a higher aldosterone concentration is required to maintain the same blood pressure. Both a kidney- or vascular mechanism appears plausible. The vascular mechanism has a precedence in the situation of patients with joint hypermobility and reduced vascular tone [77] who show increased frequency of anxiety and depression. The same conclusion of a reduced peripheral MR activity was drawn based on neuroendocrine challenge experiments with a mixed MR–GR agonist [78,79]. In line with the hypothesis of lesser MR sensitivity in depression, the adjunct administration of the MR agonist fludrocortisone led to a faster clinical response to standard antidepressants in comparison to placebo [41]. At the same time, fludrocortisone reduces aldosterone plasma levels. The clinically beneficial effect of fludrocortisone is most likely peripherally mediated, as fludrocortisone crosses the blood–brain barrier only in a very limited way [80].

A consequence of the reduced blood pressure is the activation of a compensatory increase in aldosterone release, which acts at the kidney and vascular endothelial cells to increase blood pressure in healthy subjects. Furthermore, aldosterone and other MR agonists increase salt intake via action at the CNS [81,82]. Accordingly, MR antagonists reduce appetite for salt [83,84]. Patients suffering from depression often suffer from loss of appetite, which could be caused by a subjective tastelessness of food, based on an increased salt taste threshold and increased salt preference as a consequence of aldosterone action [31].

An increase in aldosterone can be induced by salt restriction in healthy subjects. This leads to an increase in inflammation markers and metabolic markers, including triglycerides, as demonstrated in a large meta-analysis [85]. This indicates a possible causal role of increased aldosterone in metabolic and inflammatory dysregulation. Salt restriction also leads to increased signs of depression and anxiety in animal models [86,87] and in epidemiological studies [88]. In addition, serotoninergic [52,89,90] and glutamatergic mechanisms [53] play a role in the regulation of aldosterone balance.

A link to sleep exists: there is a close temporal relationship between the activity of RAAS and sleep processes; the concentration of renin and aldosterone increases in synchrony with an increase in SWS [91]. Thus, the total sleep duration as well as the duration of SWS provides a possible correlate for the activity of the MR at the NTS. In addition, conditions with sleep disturbances also show increased nightly aldosterone concentrations. These include, besides major depression, conditions like obstructive sleep apnea [92] and consequences of prolonged physical strain [93]. These observations highlight the complexity of the functional system.

## 4. Inflammation

Inflammation has long been described as a component of major depression [94]. Higher levels of inflammation are linked to treatment resistance to standard antidepressants [12,95]. In depressed patients, inflammatory changes are found including of C-reactive protein (CRP) and interleukin (IL)-6; slightly less expressed change also takes place with IL-1 [96,97]. The pathophysiology of inflammation in depression is an area of active research. The close link between autonomic nervous system and inflammation was outlined recently with the overall perspective that activation of the SNS is proinflammatory in nature and related to reduced vagal/parasympathetic function [98,99]. However, in later stages of an inflammatory process, an increase in SNS activity may be beneficial [100].

Inflammation can be experimentally induced by administration of LPS to generate a model for depression in animals [101,102,103] and humans [104]. In fact, LPS levels have been associated with some forms of depression [105], possibly based on a “leaky gut”, which may be the consequence of autonomic dysregulation. The receptor for lipopolysaccharide (LPS, endotoxin) is the toll-like receptor 4 (TLR4), which is the trigger for the activation of innate immunity. Its activation leads to the release of inflammatory cytokines via activation of the Nod-like receptor (NLR) family pyrin domain containing 3 (NLRP3) inflammasome [106,107,108].

The TLR4 is expressed in peripheral immune cells, and in astrocytes of the choroid plexus, circumventricular organs including the area postrema, and the nucleus of the solitary tract [109], as well as in the nodose ganglion of the vagus nerve [110]. This makes these TLR4 receptors accessible for systemic treatments without the need to cross the blood–brain barrier. Of note, the area postrema is intricately connected to the NTS, the primary target for aldosterone. It shows the synergism between both the RAAS and inflammation on a physiological level. This finding may have relevance for a targeted treatment approach of depression.

An increased brain expression of TLR4 has been observed in a broad range of neuropsychiatric disturbances, including major depression [111]. Of interest in the context of atypical depression is the observation that TLR4 knockout mice have less sleep need and a smaller increase in sleep rebound after sleep deprivation [112]. Brain monocytes mediate this behavior. This is in line with the increased sleep need in inflammatory conditions, possibly mediated by TLR4. TLR4 blockade may therefore lead to reduced sleep need.

TLR4 receptors are regulated by a range of relevant signal pathways, including the beta adrenergic [113]. TLR4 acts synergistically with NMDA receptor blockade to reduce white matter integrity [114] and influences the anti-inflammatory and neuroprotective role of glucocorticoids by acting on microglia [115]. Importantly, psychological stress can induce TLR4 expression. [116,117,118]. Consequently, an increase in the expression of proinflammatory factors occurs in the brain cortex. Interleukin-1beta (IL1b), cyclooxygenase 2 (COX-2), and prostaglandin E2 (PGE2) are increased and this increase is associated with an increase in depression-related behavior, as assessed in the forced swim test [118]. An activation of the nuclear factor kappa B (NFkB) inflammatory pathway was involved [119]. Intracerebroventricular administration of LPS leads to an increase in the expression of TLR4 at the ventricular ependyma, the choroid plexus and the area postrema [120], which may lead to a feed-forward cycle. Blocking TLR4 receptors pharmacologically prevented an increase in TLR4 expression within the prefrontal cortex of rats [121]. Bacterial translocation from the gut may be involved in the stress-induced TLR4 activation, as implied by a suppression of this reaction by antibiosis.

Another important inflammatory pathway, which has been related to depression, is that of kynurenine metabolism [107,122,123]. Activation of the tryptophan-degrading enzyme indolamine-2,3-dioxygenase (IDO) leads to the generation of the kynurenine, which can further be metabolized into neurotoxic compounds, like the NMDA agonist quinolinic acid. IDO is activated by LPS, i.e., the TLR4 [124,125,126], which may therefore be a common trigger of several inflammatory mechanisms. Interestingly, kynurenine can also be metabolized into the NMDA receptor- and nicotinergic alpha-7 receptor antagonist kynurenic acid, which may be neuroprotective. Of interest in our context is that aldosterone acts synergistically with inflammation inducers to increase the expression of different IDO variants in hippocampal slice cultures [127]. A proinflammatory effect of aldosterone is in line with these observations. Furthermore, tryptophan depletion, a common depression model, leads to an increase in kynurenine/kynurenic acid ratio in association with an increase in plasma aldosterone and depression-like behavior [52]. Aldosterone may be responsible for these changes, but this has not been directly demonstrated in this study.

The observed gender difference in the frequency of depression, in particular of atypical depression [16], may be linked in part to the kynurenine system. Estrogen, via the estrogen receptor 2 (ER2) may play a significant role. In a mouse model of postpartum depression, ER2 expression in the prefrontal cortex influences the expression of brain-derived neurotrophic factor (BDNF) and depression-like behavior [128]. The tryptophan-kynurenine pathway is activated by estrogen, but also influenced by progesterone (see [129]), potentially increasing the vulnerability to depression. In addition, women in the luteal phase have higher levels of aldosterone in comparison to men or women in the follicular phase [130]. This may contribute to higher inflammatory activity. ER2 activation is also associated with a reduced sensitivity of MR [131], which is in line with the increase in aldosterone plasma concentration in females. Therefore, estrogen may be an important moderator of MR activity and aldosterone regulation.

A CNS pathway has been described, which regulates peripheral inflammatory activity, i.e., the neuroimmune reflex [132,133]: inflammatory signals from the periphery are transmitted via vagal afferents to the NTS, which then led to an activation of vagal efferents to the spleen. Vagal activity suppresses inflammatory activity, creating a feedback loop. Therefore, the key anatomical target for aldosterone is involved in the regulation of inflammation. Furthermore, TLR4 is expressed in astrocytes of the area postrema and NTS and may affect the neuroimmune reflex. The involvement of astrocytes link TLR4-mediated inflammation to the glutamatergic system [134], which is the main transmitter of the afferent vagus nerve [135]. Disturbances not only of autonomic regulation, but also interoception, can be expected in case of a disruption of this pathway [67]. This may be reflected in the frequent somatoform complaints of patients with increased inflammatory activity.

Further synergism at a molecular level exists, which will be covered in the next section.

### RAAS and Inflammation

Increased levels of aldosterone are related to inflammatory and metabolic disturbances [136,137]. A correlation of increased aldosterone with an increase in CRP, insulin, total cholesterol, and triglycerides was observed in young obese adults. As mentioned already, a physiological interaction between aldosterone and inflammation exists via aldosterone’s action on the NTS. On a molecular level, aldosterone acts synergistically at the trigger for the innate immunity, the toll-like receptor 4 (TLR4) receptor [138]: co-administration of aldosterone and LPS leads to an enhanced increase in plasma, CSF, and brain interleukins, accompanied by an increase in depression-like behavior. Furthermore, TLR4 activation by lipopolysaccharide (LPS) induces aldosterone release and inflammation at the adrenal cortex that can be suppressed by an angiotensin II (ATII) receptor antagonist [139]. The inflammation-induced aldosterone release may create a feed-forward cycle. Indeed, the proinflammatory effect of ATII and its potential role in depression is well established [140].

In addition, a direct effect of aldosterone to induce inflammation has been reported: aldosterone acts on white blood cells; the neutrophil/lymphocyte ratio is correlated with aldosterone plasma concentration in patients with primary hyperaldosteronism [141]. Aldosterone induces Th17 cells [142,143], and Th17 cells are increased in the plasma of patients with depression [107].

## 5. Role of Obesity and Metabolic Parameters

Obesity has been associated with treatment resistance [144,145], and is related to atypical depression [146,147], which consistently shows lesser therapy response to standard antidepressants [3]. Obesity in depressed subjects is correlated with depression symptoms [13,148,149]. Obesity-related metabolic changes are an increase in triglyceride- and LDL levels. The increase in LDL appears to be somewhat specific to patients with atypical depression [150].

CPR and smoking predict increases in lipid levels two years later in a mixed population from the Netherlands Study of Depression and Anxiety (NESDA), which included patients with anxiety and depression [151]. This indicates that inflammation may lead to metabolic disturbances. Vice versa, altered lipid metabolism can activate the TLR4-associated NLRP3 pathway: free saturated fatty acids, like palmitate, induce inflammation in microvascular endothelial cells [152]. This led to a reduction in ZO-1/ZO-2 adhesion factors, indicating a disruption of the blood–brain barrier.

RAAS activation has a link to vascular inflammatory changes and influences inflammatory parameters in plasma [153,154]. High fat diet is associated with brain inflammation, which is mediated via activation of brain RAAS [155]. Obesity induced in rats led to vascular inflammation and is related to white matter dysfunction. This was prevented by the ATII antagonist canrenoate [156]. These associations show an integral mutual interplay between altered lipid metabolism, inflammation, and the renin–angiotensin–aldosterone system.

The question arises whether manipulation of just one of the systems can be beneficial. Regarding obesity and related lipid alterations, reduction in obesity is related to beneficial clinical outcomes, but with limitations. For example, time-restricted eating led to a reduced weight and reduced food addiction score in patients with food addiction, which was accompanied by a significant increase in brain-derived neurotrophic factor (BDNF) and a reduction in lipopolysaccharide binding protein (LBP), a co-factor for the TLR4 receptor, in the intervention vs. the control group. However, stress levels did not change after eight weeks of treatment [157]. Bariatric surgery improved depression [158], but may lead to higher levels of suicide [159]. Whereas obesity is associated with brain atrophy [160,161], reduction in weight rebalances sympathovagal activity [162] and improves brain morphology [161,163]. With weight loss, aldosterone plasma concentration was reduced in subjects who had a metabolic syndrome at baseline [137]. The latter study showed that weight loss was associated with a reduction in CRP, insulin, leptin, and sympathetic activity, as expressed by a reduced low frequency heart rate variability (HRV). Furthermore, a reduction in salt appetite, as determined by urine sodium content, was observed. The change in inflammatory markers correlated with the change in aldosterone levels. However, a large study found that an increase in BMI under treatment was associated with a better therapy response, in particular in patients with a low BMI at baseline [164]. Blood pressure was not reported in this study. It is not clear if the latter finding is related to an increase in appetite, as expected with the improvement of melancholic symptoms, but the association of clinical improvement with weight gain shows that increased weight is not universally associated with higher depression severity. As always, the context of the determined biomarkers is important.

The combination of these characteristics: increased inflammation, obesity and metabolic abnormalities, autonomic dysregulation based on hyperaldosteronism and consequently central MR activation, as well as signs of atypical depression could constitute a specific biological subtype with relevance for a targeted treatment approach.

## 6. Clinical Relevance

We described a form of depression, which shows signs of metabolic, inflammatory, and neuroendocrine alterations, including a lower level of cortisol and a potentially higher level of aldosterone. This form in unfortunately characterized as “atypical”, which is true when compared to the textbook form of depression, which often characterizes melancholia. Importantly and often overlooked, this form of depression is prevalent in females and, in particular, young females, who show atypical features in about 80% [4,16,165]. In addition, these patients show an earlier age of onset in comparison to non-atypical patients, signs of somatization, and an increased risk of drug abuse [166]. This is a crucial group of patients, which do not frequently respond to the first line psychopharmacological treatment option of SSRIs. There is evidence that they respond to compounds, which increase noradrenergic and dopaminergic activity, in particular monoamine oxidase inhibitors [167] and interestingly St. John’s Wort [168,169], both of which increase dopaminergic activity [134,170]. This is of interest as dopaminergic dysfunction appears to be a primary consequence of inflammation. Treatment with l-dopa, i.e., and increase in dopaminergic activity, was beneficial in these subjects [171]. Complementary to this inflammation-induced mechanism, aldosterone has a direct effect on dopaminergic transmission [44,87,172]. Importantly, the dopaminergic deficiency can be regarded as a consequence of high inflammation and high aldosterone levels. This may lead to a further understanding of this disorder as a dopamine deficiency disorder, besides other characteristics.

How immune activation affects brain activity is nevertheless not entirely clear. As mentioned, a role of the autonomic nervous system is suggested by the observation that autonomic changes are associated with metabolic alterations: lower HRV in patients with depression as a sign of reduced vagal activity is associated with metabolic disturbances [173]. Similarly, cardiac physiologic markers, including heart rate, low cardiac autonomic balance, and low RSA, were associated with abnormal metabolic markers [174] and even predicted the development of metabolic abnormalities, including lower HDL [14]. In addition, an increase in sympathetic activation is associated with lipolysis and an increase in triglycerides [175]. This may point to autonomic dysfunction as the primary disturbance. In this context, the role of vagal activity is of importance and complex: the sensory (afferent) vagus is involved in sensing satiety [176], which may be interrupted in subjects with hyperphagia.

Regarding efferent vagal pathways, high levels of central obesity, as expressed by waist circumference, is consistently related to low HRV as a potential sign of low vagal activity [177]. Low HRV and low baroreflex sensitivity are also associated with higher liver fat content in patients with early type II diabetes [178]. Accordingly, weight loss leads to an increase in HRV [162]. Vice versa, recent meta-analysis of auricular vagus nerve stimulation shows beneficial effects on glucose tolerance and BMI, but the relevance of the effect size was questioned [179]. However, other studies show that vagus nerve blockade in obese subjects [180] or vagotomy in patients with obesity [181] and in mice [182] improves metabolic function. The latter observation makes intuitive sense, as the vagus nerve is regarded as a “rest and digest” nerve. Metabolic parameters are in fact regulated by the autonomic nervous system in complex ways [183]. The role of increased vagal activity in health conditions requires further study [184].

Regarding the question which of these systems should be primarily targeted to achieve a clinical benefit, a recent study showed that a reduction in lipids and inflammation may not be sufficient to improve the underlying pathophysiology; reduction in these markers with a statin as an adjunct treatment to standard antidepressants in comparison to placebo in patients with depression did not have any effect on clinical outcome of depression [185]. These findings imply that the root cause of the disorder is upstream of inflammation and lipid dysregulation. One may consider the role of the autonomic nervous system [184].

## 7. Brain Imaging Studies

Characteristics which define a subtype with immuno-metabolic changes are an increase in BMI or triglyceride levels and an increase in inflammation markers. In extension, this may be associated with a reduced cortisol and increased plasma aldosterone concentration and autonomic dysfunction. Do these markers have consequences for brain morphology? We demonstrated that an increase in BMI and triglycerides is associated with an increase in the volume of the lateral ventricles of the brain [186,187], and that ventricular volumes at baseline were related to treatment outcome. Inflammation has been demonstrated as a potential cause for ventricular volume enlargement, for example, in patients with multiple sclerosis [188]. More specifically, the relationship between immuno-metabolic forms of depression and increased ventricular volume was confirmed in a large observational study [189]. An increase in ventricular volume was attributed to an increase in the volume and increased activity of the choroid plexus [188]. The association between increased choroid plexus volume and inflammatory markers has been confirmed in patients with bipolar disorder [190,191] and major depression [192].

Alterations in gene expression of the choroid plexus have been observed in patients with depression, in particular of markers of transforming growth factor beta (TGFbeta) [193]. As patients with primary hyperaldosteronism have reduced TGFbeta1 plasma levels, which could be reversed with spironolactone [194], a causal link may exist. In our studies, an additional feature occurred: we found an association between enlarged ventricles and higher volume of the choroid plexi with a reduction in anterior and medial parts of the corpus callosum [186,187], confirming earlier observations [195]. How these changes are mediated is not immediately clear. We proposed a model, in which the change in the choroid plexus is a consequence of autonomic dysregulation, which leads to an increase in CSF release and potentially a compression of parts of the corpus callosum [196]. Therefore, balancing the autonomic nervous system may be a priority to improve these conditions.

## 8. The Role of Childhood Trauma

Childhood trauma has been associated with neuroendocrine characteristics, in particular hypocortisolism, inflammation, and autonomic alterations, with a link to orthostatic dysregulation and baroreceptor dysfunction [197]. Furthermore, high aldosterone levels and low blood pressure occur in subjects with a history of childhood trauma [198]. Traumatization affects brain morphology: higher volumes of the brain ventricles and smaller volumes of the corpus callosum in comparison to control subjects were observed in traumatized children who developed PTSD [199,200]. Accordingly, and in line with our studies, white matter integrity of the corpus callosum as measured by fractional anisotropy (FA) was reduced in bipolar subjects with higher levels of childhood trauma and higher levels of inflammation [201]. This study is based on the same sample which showed larger choroid plexus volumes with inflammation [191]. This observation has parallels to morphological changes in patients with depression less responsive to therapy [186,187]. It further supports the association between a history of childhood trauma and risk of antidepressant treatment refractoriness [202,203].

## 9. Possible Therapeutic Interventions

To understand the plausibility of a targeted pharmacological intervention it is useful to understand its basic physiology. A broadly studied intervention is physical exercise as a treatment of depression [204]. There is preliminary evidence that atypical depression is particularly sensitive to exercise [205]. Furthermore, the combination of exercise to sertraline increases parameters of HRV, which was regarded as an increase in vagal tone [206]. In our context, it is of interest to note that exercise does not only reduce weight, but also reduces ATII, aldosterone, and norepinephrine levels in the general population [207]. This has not been studied in patients with depression. As already reported, weight loss induced by Bariatric surgery is associated with a reduction in aldosterone concentration [208], normalization of brain morphology [209], and improvement of depressive symptoms. This supports the role of aldosterone in the characterized forms of depression.

Regarding pharmacological interventions, the MR agonist fludrocortisone showed some beneficial effect in speeding up recovery in patients with depression [41]. Interestingly, fludrocortisone leads to a suppression of aldosterone release via a feedback mechanism, which also involves an increase in blood pressure. Alternatively, ATII antagonists have shown benefits, primarily in open label studies (see [44]). A limitation of this approach is that ATII antagonists not only reduce aldosterone levels, but also blood pressure, which may counteract its efficacy, as both signals work in opposition at the NTS.

### 9.1. Glycyrrhizin/Enoxolone

Glycyrrhizin and enoxolone have been widely studied for their pharmacological and biological properties [210]. Their pharmacokinetic and toxicological profile are well known. As glycyrrhizin is converted into enoxolone in the gut, and only enoxolone is taken up to act systemically, we regard these compounds as equivalent and do not differentiate their actions in the following [211]. A related compound is a modified enoxolone, carbenoxolone, which is discussed in the same way. The main areas of interest are the effect of enoxolone to inhibit 11betaHSD2 and to antagonize TLR4 receptors.

#### 9.1.1. Glycyrrhizin/Enoxolone Reduces Aldosterone via Inhibition of the 11betaHSD2

An approach to reduce aldosterone plasma concentrations and avoid the blood-pressure-reducing effects of ATII antagonists or ACE inhibitors, can be achieved with glycyrrhizin [212], from the licorice plant, glycyrrhiza glabra, and its active metabolite glycyrrhetinic acid (enoxolone) (Figure 1). Their mechanism of action features the inhibition of the 11betaHSD2 at the kidney and vascular endothelial cells.

11betaHSD2 is the enzyme which protects MR in the NTS and the kidney from being occupied by cortisol/corticosterone by metabolizing cortisol into the inactive cortisone. This allows aldosterone to compete. With the inhibition of this enzyme, cortisol can bind to kidney and vascular endothelial MR. This leads to an increased uptake of sodium and volume and a reduction in potassium, together with an increase in blood pressure. This effect is classified as pseudo-hyperaldosteronism in pathological conditions [210]. As a consequence of the MR stimulatory effect of cortisol, a feedback mechanism acts to reduce aldosterone release [214,215,216,217]. The classical perspective is that renin is primarily suppressed by an increase in blood pressure or plasma sodium concentration at the juxtaglomerular apparatus [218], but a direct effect at a kidney MR to suppress renin was also suggested [219]. In short, the consequences of renal and vascular MR activation by cortisol lead to a compensatory reduction in aldosterone release (Figure 2a).

The dose range is critical for its effect: enoxolone acts in a reversible fashion in a dose of up to 500 mg daily, but may be irreversible in a dose of 1500 mg daily [220]. On the basis of these parameters, a non-effect dose of approx. 100 mg daily has been described [221,222]. However, a crossover study in healthy subjects with a dose as low as 100 mg glycyrrhizin demonstrated a reduction in renin and aldosterone and to slightly increase blood pressure [223]. The blood-pressure-increasing effect of 11betaHSD2 inhibition is in part due to a direct action at vascular MR leading to a vascular contraction. This involves an increased sensitivity to norepinephrine and cortisol [224], as shown in healthy subjects. The vascular effect of glycyrrhizin from licorice (dose 290–370 mg for two weeks) is reflected by an increase in aortic and popliteal pulse wave velocity, alongside the increased blood pressure and a reduced orthostatic response [225]. In addition, this study showed a reduction in low frequency heart rate variability, potentially as a sign of reduced sympathetic activity.

A direct CNS effect of glycyrrhizin/enoxolone also exists: the administration of glycyrrhizin intracerebroventricularly in rats led to an activation of the hypothalamic paraventricular nucleus, which goes along with an increase in blood pressure and renal sympathetic activity [76] by direct sensitization of the PVN neurons to corticosterone. A similar mechanism can be proposed for a direct action of enoxolone at the NTS, which would lead to higher MR activation. This may induce an unwanted MR stimulation at the NTS with higher doses of enoxolone. It is therefore important to keep in mind that the rationale for the use of glycyrrhizin/enoxolone is to reduce aldosterone peripherally and therefore reduce MR activation at the NTS. This targeted approach can be achieved as enoxolone crosses the blood–brain barrier to only a limited extent [226].

It is of interest to note that there is a potential alternative mechanism for enoxolone and the related compound carbenoxolone to affect hypothalamic activity: carbenoxolone administration into the ventromedial hypothalamus reduced hypothalamic theta oscillations in a sodium-channel-dependent way. Mediation by gap junction block was suggested [227]. Involvement of gap junction blockade was also proposed for the effect of carbenoxolone to block vagally mediated hippocampal theta oscillations [228]. However, these effects are unlikely to be relevant in the proposed doses due to the limited BBB permeability.

#### 9.1.2. Glycyrrhizin Reduces Inflammation via TLR4 Inhibition

Glycyrrhizin/enoxolone have been recognized as neuroprotective by inhibiting inflammation via TLR4 antagonism and reduction in high mobility group box-1 (HMGB-1) protein and action at its receptor RAGE (receptor for advanced glycation endproducts) [229,230,231,232,233,234,235,236,237,238].

Glycyrrhizin reduces the generation of oxidative stress from neutrophils [239]. In a hamster model, an extract from glycyrrhiza glabra had anti-inflammatory effects against COVID infection, which reflects direct inhibitory action at neutrophils and reduced oxidative stress [240]. A suppression of the differentiation of Th1, Th2, and Th17 cells was involved in the action of glycyrrhizin in this study. A direct effect of glycyrrhizin on T-cells was confirmed in another experiment [241], with a focus on Th17 T-cell differentiation and reduction in IL-17 production [240,242]. Overall, glycyrrhizin/enoxolone target immune cells. In addition, and complementary to this effect, enoxolone protects against oxidative stress in intestinal epithelial cells by activating the phosphoinositide 3-kinase/protein kinase B (PI3k/Akt) pathway [243]. Similarly, advanced glycation endproducts (AGEs) induce oxidative stress and inflammation in umbilical cord epithelial cells. This was also inhibited by glycyrrhizin [244].

A broadly discussed inflammatory mechanism in depression is related to the kynurenine pathway related to the activation of the tryptophan-metabolizing enzyme IDO, as described above. This pathway is activated via TLR4 receptor activation. Consequently, glycyrrhizin/enoxolone dampens this pathway, as demonstrated in an unpredictable stress model of depression [245]. This dampening effect went along with a reduction in depression-like behavior. In addition, glycyrrhizin and glycyrrhetinic acid block kynurenine aminotransferase 2 (KAT2), which metabolizes kynurenine to kynurenic acid, an antagonist at the NMDA and alpha-7 nicotinergic receptor [246].

An indirect functional immune suppression also exists, as described above. As the reduction in aldosterone and inhibition of TLR4 affect the activity of the NTS, these actions may affect the neuroimmune reflex. The neuroimmune reflex [132,133] establishes a feedback mechanism to suppress increased peripheral inflammation, i.e., can be regarded to ensue homeostasis. The key anatomical structures are the vagus afferents, connecting to the vagus nerve to the NTS proper, and the efferent vagus, projecting from the nucleus ambiguous and dorsal vagal nucleus into the periphery. These efferents to the spleen suppress immune response, as determined by interleukin release, via cholinergic alpha-7 nicotinergic receptors (See Figure 3). This association between the central anatomical element of the anti-inflammatory reflex, the NTS, with the area postrema, leads to an additional important functional connection relevant for the anti-inflammatory effect of enoxolone: TLR4 is expressed in astrocytes of the area postrema and NTS. This demonstrates an intriguing synergism between the main effects of enoxolone affecting NTS activity: firstly, by targeting astrocytes at the NTS via TLR4 inhibition and, secondly, targeting MR/11betaHSD2 co-expressing cells in the NTS via a reduction in aldosterone plasma concentration.

#### 9.1.3. Behavioral Effects of Glycyrrhizin/Enoxolone

An animal model (single prolonged stress) leads to an induction TLR4 and HMGB1 expression in the basolateral amygdala, which was accompanied by neuroinflammation. These changes could be inhibited by glycyrrhizin [116,117]. Glycyrrhizin counteracts depression-like behavior in mice administered with LPS, the ligand of the TLR4 receptor [235]. Glycyrrhizin blocks the effect of subchronic restraint stress: this form of stress induction inhibits locomotor activity, which could be reversed by glycyrrhizin [247]. The behavioral effect was accompanied by an increase in corticosterone release. Interestingly, the beneficial behavioral effect was observable for rats treated for 10 days, but not for several weeks. Glycyrrhizin reduced anxiety- and depression-like behavior in a neuropathic pain model (sciatic nerve ligation), accompanied by a reduction in prefrontal cortex microglia activation by blocking HMGB1 action [229]. Enoxolone in a dose between 10 mg/day and 50 mg/day administered into the stomach of rats that underwent chronic unpredictable stress reduced depression-like behavior in a dose-dependent way. This included a reduction in immobility in the forced swim test and a reversal of a reduced sucrose intake. This was accompanied by a reduction in HPA axis activity, a reduction in the stress-induced increase in liver enzymes and interleukins, and was associated with increased prefrontal and hippocampal serotonin and norepinephrine concentrations. Furthermore, prefrontal BDNF concentration increased [248]. A further study confirmed the role of BDNF and its receptor tropomyosin receptor kinase B (TrkB) in the antidepressant response of glycyrrhetinic acid in a chronic social defeat model. That study reported the involvement of the omega-3 fatty acid docosahexaenoic acid (DHA) as a mediator [249]. In another experiment, glycyrrhizin, an active component of an extract, reduced salt appetite when administered orally in a dose of 150 mg/kg per day for two weeks. The extract had a concentration of glycyrrhizin of approx. 7% [250]. The reduced salt appetite is a sign of reduced central MR activation, as intended. This was associated with an anxiolytic effect and an increase in corticosterone release. The increase in corticosterone was observed in unstressed, not stressed rats [251]. A non-significant reduction in aldosterone occurred, which indicates a reduction in the aldosterone/corticosterone ratio.

An association of the response to glycyrrhetinic acid and change in brain structure has been described: chronic social stress leads to morphological changes in selected brain regions, including white and gray matter areas, and to an alteration in diffusion tensor imaging parameters, including a reduction in fractional anisotropy. These changes could be reversed with the administration of glycyrrhetinic acid [252]. This is interesting in the context of findings that the aldosterone/cortisol ratio as well as inflammatory activity is related to similar brain morphological changes (see [196]).

An inhibition of the NFkB-IL-17-pathway by glycyrrhizin is associated with brain protection in an ischemia model, which was accompanied by a reduction in neurological deficits [236,253]. This is relevant, as brain morphological changes may be mediated by this inflammatory pathway. Action at the choroid plexus structures appear to be involved in the effect of IL-17 and may mediate depression as IL-17 is associated with choroid plexus volume in patients with bipolar disorder [190]. The choroid plexus appears to be an entry point of IL-17-producing Th-17 T-helper cells, which play a key role in demyelination [254]. We are currently researching whether enoxolone leads to changes in brain morphology in patients with depression, and for which patients this may be beneficial. For details on the objectives and design please see (https://www.clinicaltrials.gov/study/NCT05570110) (accessed on 5 October 2025).

In addition to the direct immunosuppressive effect on brain structures, the same inflammatory pathway is relevant for the beneficial behavioral effect of glycyrrhizin via improved integrity of the gut [242,255]. In line with this, with these observations, a glycyrrhizin-containing extract downregulated the expression of angiotensin-converting enzyme 2 (ACE2) in the gut [251]. The ACE2 downregulation was observed in the same study, which reported reduced anxiety-related behavior [250], as reported before. The reduction in ACE2 expression could be due to reduced inflammatory stress [256]; however, this has not been demonstrated directly.

#### 9.1.4. Effect of Glycyrrhizin/Enoxolone on Metabolic Parameters

Glycyrrhizin demonstrated an effect on metabolic parameters: in an animal model (doxorubine-induced cardiomyopathy), administration of glycyrrhizin led to a decrease in triglyceride levels, whereas high density lipoprotein (HDL) levels increased [257]. In addition, an increase in blood pressure was observed in this study. Similarly, four-week treatment with an ethanolic extract of licorice led to a reduction in triglycerides; plasma glucose and weight gain in high fat diet (for 10 weeks) induced metabolic disturbance in rats. This was accompanied by changes in gut flora, liver steatosis, and signs of reduced liver inflammation [258]. Furthermore, oral administration of enoxolone led to an improved lipid profile and a reduced weight gain in a high fat diet model [259]. The effect on lipid metabolism and inflammation may involve the stabilization of ACE2 in the liver [260]. It may also involve a change in sympathetic hepatic nerve activity, which could be mediated via the gap junction inhibitory effect of these compounds, as described for carbenoxolone [261]. Finally, metabolic changes in obesity include higher levels of free saturated fatty acids, which induce inflammation, and a reduction in ZO-1/ZO-2 adhesion factors in microvascular endothelial cells. These changes could be reversed with the administration of glycyrrhizin [152].

#### 9.1.5. Glycyrrhizin Reverses Catecholamine Depletion—Autonomic Activity as Primary Driver?

Back to the starting point focusing on catecholamines and autonomic regulation, Schildkraut’s framework of the catecholamine deficiency hypothesis of depression may best apply to patients with fatigue and/or low blood pressure. The overlap between these symptoms is well documented [206,262]. Ammonium glycyrrhizinate (100 and 150 mg/kg) intraperitoneally (i.p.) has demonstrated a reversal of behavioral changes induced by reserpine [263], potentially mediated by increased monoamine levels. Another study reported an antidepressant effect of glycyrrhizin which could be inhibited by antagonism at noradrenergic and dopaminergic receptors: reduced immobility was observed in the forced swim at a dose of 150 mg/kg extract containing approx. 8% glycyrrhizin. Doses of 75 mg/kg or 300 mg/kg were not effective, supporting an optimal dose range. The effect could be blocked by sulpiride (D2 antagonist) or prazosin (alpha1 antagonist) [264]. In a further experiment, utilizing a norepinephrine depletion model with fusaric acid, a dose-dependent reversal of anxiety-related behavior was observed with the administration of glycyrrhizin, which was also accompanied by an increase in brain metabolic signs (increase in adenosine triphosphate (ATP)) and integrity (increase in brain-derived neurotrophic factor, BDNF) [265]. Doses of 200 mg/kg and 300 mg/kg orally (dissolved in distilled water), but not 100 mg/kg, were effective on these parameters.

An extended overview of the effect of enoxolone is depicted in Figure 3.

**Figure 3 pharmaceuticals-18-01517-f003:**
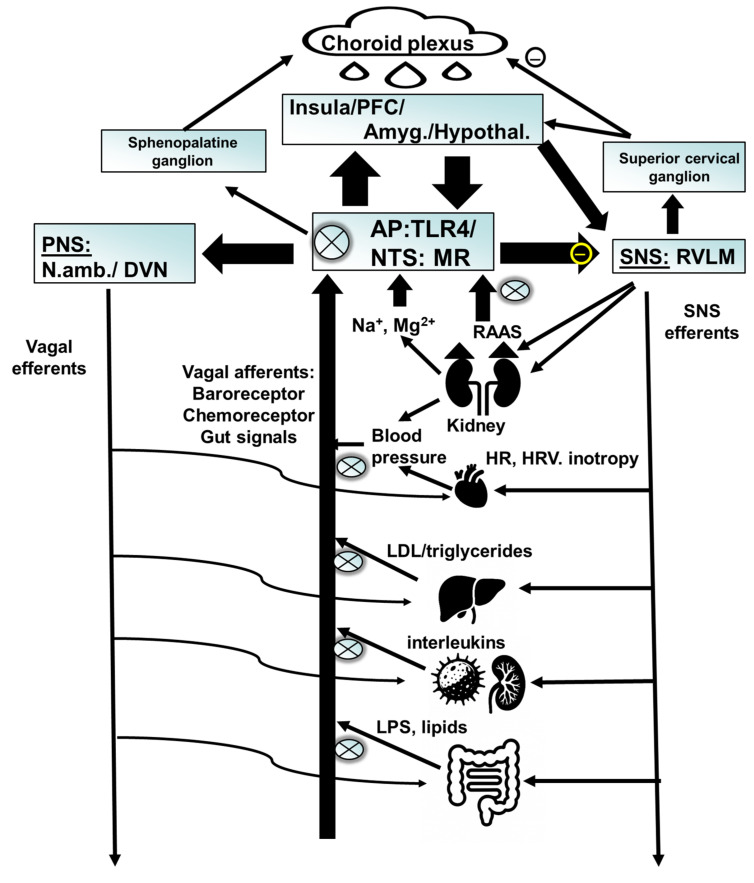
**Schematic summary of the underlying neurobiological targets of enoxolone:** 1. Enoxolone inhibits the release of aldosterone from the adrenal cortex due to the fact that it inhibits the kidney and endothelial 11betaHSD2. This allows cortisol to bind to peripheral MR, which via feedback inhibits renin release, the generation of angiotensin, and finally the activation of ATII-mediated aldosterone release. The reduction in aldosterone reverses the activation of NTS neurons, normalizes baroreceptor reflex and other inputs from the afferent vagus. Consequently, efferent vagus activity is increased and SNS activity is decreased. This has effects on multiple levels: improvement of cardiovascular regulation, sensitization of the neuroimmune reflex, thereby reducing inflammatory activity and improvement of metabolic activity. Via its link to the hypothalamic paraventricular nucleus, it may affect HPA axis and RAAS activity indirectly. Through an effect on the SNS, kidney- and adrenal function are modulated. 2. Enoxolone also inhibits TLR4 activity, with direct consequences for immune cells. This affects the neuroimmune reflex via targeting the vagal nodose ganglion. Finally, it reduces inflammation of the gut. These signals feedback via the afferent vagus nerve or humoral mechanism to influence the activity of the NTS. PFC: prefrontal cortex; Amyg: amygdala; hypothal.: hypothalamus; NTS: nucleus of the solitary tract; AP: area postrema; PNS: parasympathetic nervous system; SNS: sympathetic nervous system; N.amb.: nucleus ambiguous; DRN: dorsal vagal nucleus; RVLM: rostroventerolateral medulla; HR: heart rate; HRV: heart rate variability; RAAS: renin–angiotensin–aldosterone system; LDL: low density lipoprotein. MR: mineralocorticoid receptor; TLR4: toll-like receptor 4; LPS: lipopolysaccharide (endotoxin). X: target for enoxolone.

#### 9.1.6. Clinical Studies

How do the reported biological activities of glycyrrhizin translate into clinical outcome? Beneficial effects in animal models of depression and anxiety were reported above. The number of clinical observations is, however, limited. Glycyrrhizin is a common ingredient in Japanese, Chinese, and Ayurvedic medicine, for example, of the Japanese Yokukansan, and showed beneficial effects on depressive symptoms in a placebo-controlled trial [266,267]. It is, however, difficult to attribute this to any specific ingredient. In clinical studies in patients with depression, glycyrrhizin as a component of an extract from glycyrrhiza glabra demonstrated antidepressant activity in comparison to a historic control [212]. The effect was most pronounced in a patient group with a low vs. higher systolic blood pressure (median split at 127 mmHg). Clinical improvement in the glycyrrhizin-treated subjects was associated with a reduced heart rate and a shortening of sleep duration. Another study explored the anti-inflammatory effect of glycyrrhizin. In a randomized placebo-controlled trial, adjunct treatment with glycyrrhizin to an SSRI led to an overall statistically significant benefit of glycyrrhizin vs. placebo of depression symptoms. Interestingly, this effect was restricted to subjects with higher baseline levels of C-reactive protein (>3 mg/dL), i.e., a sign of systemic inflammation [268]. The subjects of this study were primarily male (89%) and had on average a normal BMI of approx. 23.4 kg/m^2^, i.e., they did not show obvious metabolic abnormalities. Besides a significant clinical improvement, the actively treated subjects showed a reduction in TNFalpha and IL1beta, which was significantly different from the placebo group. In addition, an interim analysis from an ongoing randomized placebo-controlled study with enoxolone, 100 mg vs. placebo, in hospitalized patients with depression [269], confirmed target engagement by a significant reduction in the night urine aldosterone/cortisol ratio and of plasma CRP. Treatment responders showed preferentially higher CRP levels at baseline as well as a history of more childhood abuse. Put together, we provided evidence for an antidepressant effect of glycyrrhizin/enoxolone in combination with standard antidepressants in a population, which can be identified based on biological (inflammation, autonomic disturbances, high aldosterone/cortisol levels) and anamnestic (childhood trauma) parameters. We hypothesize that the assessments of these biomarkers can be performed BEFORE any standard antidepressant is applied, which may avoid the clinical expression of treatment refractoriness. These subjects can be regarded as treatment refractory based on their underlying biology. Therefore, enoxolone/glycyrrhizin could be a first line treatment for subjects with elevated risk of treatment resistance.

#### 9.1.7. Safety

Glycyrrhizin occurs in a concentration of about 0.2% in licorice confectionery. Thus, 100 g of licorice confectionery contains about 200 mg glycyrrhizin. The US Food and Drug Administration (FDA) published the latest guideline in 2017. In it, glycyrrhizin-containing foods are classified as GRAS (generally regarded as safe), provided that concentration limitations are observed [270]. The scientific literature considers 200 mg glycyrrhizin (equimolar to 114 mg enoxolone) as safe [210]. This recommendation is supported by original work characterizing the effect of glycyrrhizin on blood pressure, electrolytes, and endocrine changes [214,222]. In this context, it is important to note that possible adverse effects of higher or prolonged dosing can be easily monitored and may lead to a dose adjustment. These are the systolic blood pressure and plasma electrolyte concentrations. Since enoxolone is the active systemic metabolite of glycyrrhizin, the effects described for glycyrrhizin are also relevant for enoxolone.

Several studies with enoxolone in humans should be mentioned here, which underline the safety of enoxolone:

(1) A single dose of 500 mg, 1000 mg, or 1500 mg of enoxolone was administered to six healthy volunteers each to determine the pharmacokinetics of the substance. With increasing dose, an increasing blockade of the 11betaHSD2 enzyme was shown, expressed by an increase in the plasma cortisol/cortisone ratio. While a dose of 1500 mg may lead to a prolonged blockade of the enzyme, this was not observed at a dose of 500 mg at which the blockade was reversible. Good tolerability was observed, and no side effects were reported. Blood pressure and pulse were not altered (measured up to the time of 10 h after administration) [220]. (2) Enoxolone at a dose of 500 mg was administered to healthy volunteers for eight days. The expected change in cortisol/cortisone concentration ratio occurred. Two subjects complained of headaches during the period of ingestion and five subjects developed mild symptoms, one subject, ankle edema [271]. (3) In a crossover study, sixteen healthy subjects were given a dose of 130 mg of enoxolone over a period of five days. The expected change in cortisol/cortisone ratio was found. Pharmacokinetically, a slight accumulation of enoxolone was observed. The endocrine effects were reversible after four days at the latest [272]. (4) In a prospective double-blind crossover study, seven patients requiring hemodialysis with anuria were given a dose of 1000 mg over two weeks. The cortisol/cortisone ratio changed as expected; no increase in blood pressure was observed, but a decrease in plasma potassium. No other side effects have been reported [273]. (5) A total of 647 patients were randomized to a study (1:1) after surgery to remove a pituitary adenoma and followed for up to five years. In total, 512 patients completed the study, of which 268 were treated with enoxolone. A total of 135 patients completed the study prematurely, with no difference in the enoxolone vs. the placebo arm. After 1 month and 6 months of treatment, there was evidence of improvement in cognition with enoxolone- vs. placebo-treated patients [274]. (6) Ten chronic hemodialysis patients were included in a placebo-controlled trial and treated with either 500 mg of enoxolone or placebo for 12 weeks. Nine out of ten patients showed the desired reduction in plasma potassium concentration. Overall good tolerability has been reported [275].

## 10. Limitation

The current report of the effects of glycyrrhizin/enoxolone focus on biological properties related to a specific form of depression, which is less responsive to standard antidepressants. This form shows signs of inflammation, increased aldosterone levels, low cortisol, and low blood pressure. The motivation to study these compounds was to find an intervention which reverses these biomarker characteristics and the underlying neurobiology, therefore improving clinical outcome. We reported aspects of enoxolone, which are supportive of this concept. Other properties have not been discussed in detail. These include the antiviral effect of these compounds [276], anticancer, liver protective and SARS-Cov-2-targeting mechanisms [210,277].

The molecular mechanisms of interest were specifically the 11betaHSD2-inhibitory effect, the TLR4 and the HMGB1 antagonistic effect. Other molecular targets of enoxolone include its effect to inhibit 11betaHSD1 [278], inhibit cytosolic 5 beta-reductase and microsomal 3 beta-hydroxysteroid dehydrogenases [279], inhibit 15-hydroxyprostaglandin dehydrogenase and delta 13-prostaglandin reductase [280], and potentially other steroid-metabolizing enzymes. Enoxolone appears to compete with low affinity with aldosterone at the kidney MR [281]; regulates connexin 43 hemichannels [282]; and it can block cardiac NaV1.5 channels [283] and T-cell Kv1.3 channels [284]. Furthermore, glycyrrhetinic acid appears to inhibit p-glycoprotein [285], which is also involved in the transport of steroids. It is furthermore of importance that glycyrrhetinic acid has several active metabolites [210,286,287,288], which cannot be addressed here. Furthermore, the broad functional consequences, particularly the reduction in aldosterone concentration, the increase in cortisol concentration, and the shift in autonomic balance have far-reaching indirect effects. These changes may explain the metabolic consequences of glycyrrhizin/enoxolone administration and contribute to modifications in the brain–gut axis. More work needs to be carried out to disentangle these effects.

## 11. Conclusions

We identified biomarkers related to treatment resistance to standard antidepressants in depression. These include autonomic (low blood pressure, low heart rate variability), inflammation (higher CRP), metabolic (higher triglycerides and higher BMI), and neuroendocrine (high aldosterone/cortisol ratio) markers. The underlying neurobiology points to a dysfunction of peripheral MR activity, an increase in aldosterone plasma concentration and associated autonomic dysregulation mediated by an action of aldosterone on the nucleus of the solitary tract. This and a synergistic mechanism via increased expression and activation of the TLR4 is involved in the activation of the innate immune system in some forms of depression. Glycyrrhizin/enoxolone has specific properties, primarily to inhibit the 11betaHSD2 and block the TLR4, to overcome the underlying neurobiology. The clinical effect in a specific subgroup is supported by data from placebo-controlled clinical trials. Additional studies are ongoing to further test these findings.

## Figures and Tables

**Figure 1 pharmaceuticals-18-01517-f001:**
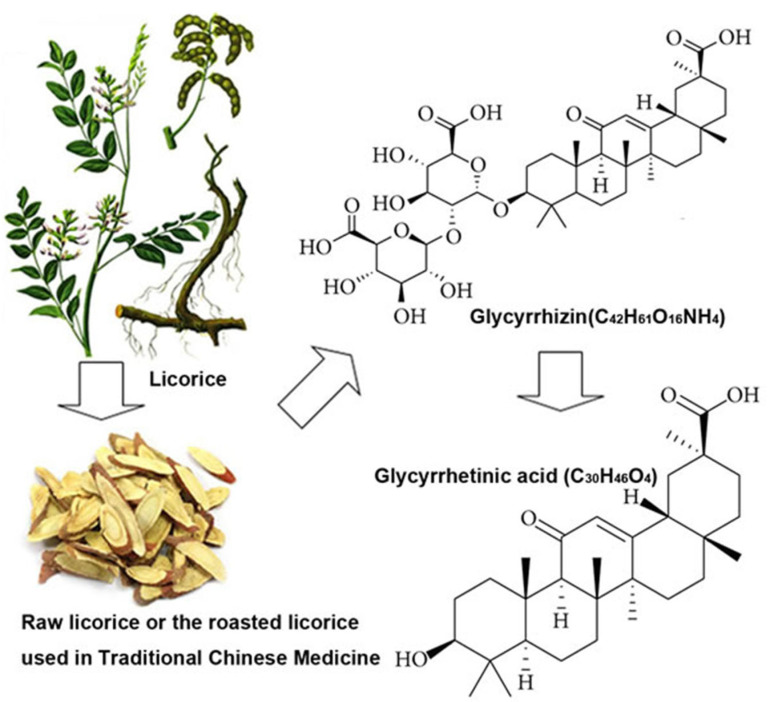
**Source of glycyrrhizin and enoxolone:** Glycyrrhizin is derived from glycyrrhiza glabra. Glycyrrhetinic acid (enoxolone) is the active metabolite of glycyrrhizin. It is generated in the gut by hydrolysis by gut bacteria (from [213]).

**Figure 2 pharmaceuticals-18-01517-f002:**
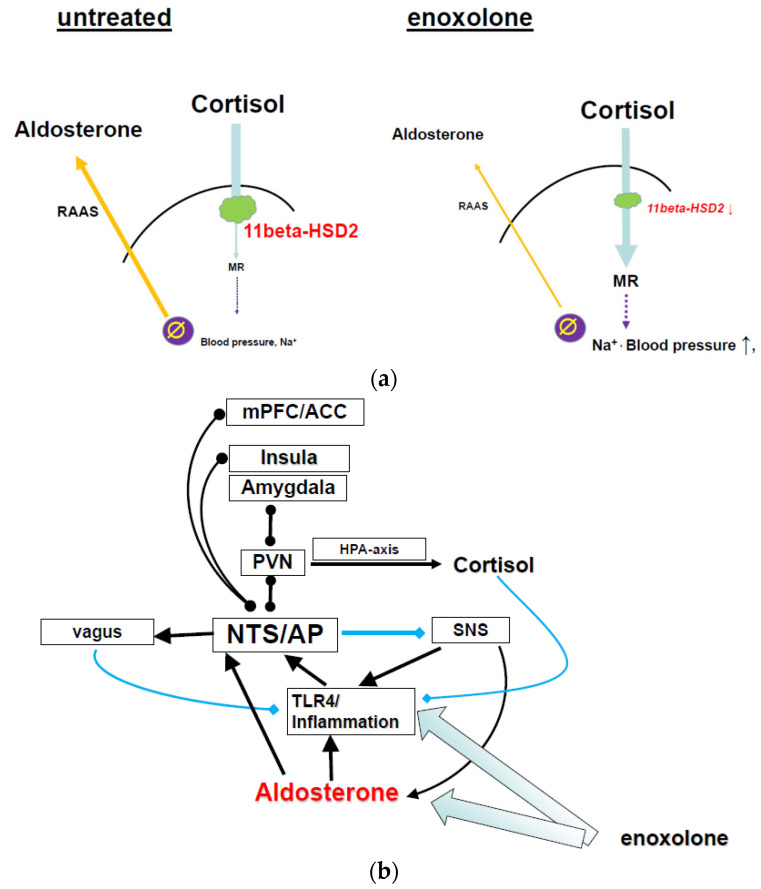
(**a**) **Mechanism of enoxolone to reduce aldosterone:** Cortisol is an MR agonist, similar to aldosterone, but has a much higher (500-fold) plasma concentration. To allow aldosterone to bind in aldosterone sensitive cells, like the kidney, the enzyme 11betaHSD2 is expressed, which degrades cortisol quickly (left). Enoxolone inhibits this enzyme, allowing a higher concentration of cortisol to activate MR in the kidney and vascular epithelial cells. This leads to an increase in sodium and water reuptake and an increase in blood pressure, which inhibits renin release. There is some support for direct effect of MR activation on the kidney to suppress renin release. The reduction in aldosterone is the step which mediates the CNS effects (**b**). Note: the size of the font and symbols represents the activity/concentration of the depicted parameters. (**b**) **Schematic effect of the effects of enoxolone:** The reduction in aldosterone, induced by enoxolone, has two synergistic, but independent effects: aldosterone activates the nucleus of the solitary tract (NTS), which is a key anatomical area to regulate the balance between sympathetic and parasympathetic (vagal) activity. The NTS has bi-directional connections to higher brain areas, including prefrontal cortical areas and the insula, and is therefore involved in interoception and affect regulation. A parallel pathway is related to the molecular synergism between MR and TLR4 activation, which is proinflammatory, and also affects NTS activity. NTS activity feeds back via vagal efferents to reduce inflammation and to reduce SNS activity, supporting an anti-inflammatory effect. In addition, enoxolone has a direct TLR4 antagonistic effect, which reduces inflammatory activity even further (see below). Arrows depict activating pathways, cubes inhibitory.

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
