# Peer review of "Discovery of Personalized Treatment for Immuno-Metabolic Depression—Focus on 11beta Hydroxysteroid Dehydrogenase Type 2 (11betaHSD2) and Toll-like Receptor 4 (TLR4) Inhibition with Enoxolone"

_pharmaceuticals, 2025, doi:10.3390/ph18101517_

Round 1

Reviewer 1 Report

Comments and Suggestions for Authors

The article entitled “Discovery of personalized treatment for immune-metabolic de pression – focus on 11beta hydroxysteroid dehydrogenase type 2 (11betaHSD2) and toll like receptor 4 (TLR4) inhibition with enoxolone” written by Harald Murck. The overall manuscript has generated some useful information. However, to improve the quality of manuscript and to help readers following are the comments need to be addressed. 

  1. Abstract part needs to be modified. It needs to be short, clear and to the point avoiding the extra details in section.
  2. Check the journal rules for the number of keywords in the review article.
  3. Abbreviated words such as HPA, STAR-D, HRV and many others needs to be fully pronounced at least for one time.
  4. Introduction part also need modification and linking one paragraph to other including clear objectivities of the study need to be focused. The following article will help in this regard. The author needs to corelate and include these. Actas Esp Psiquiatr 2024;52(1):28-36, ISSN:1578-2735, doi: 10.62641/aep.v52i5.1748, doi: 10.34133/research.0537.
  5. The following lines have some sentence or grammatical error leading to confusion in reader’s mind.
  • ‘The fact that patients with depression have rather a low vs. high blood pressure is well established [21,22]’ is confusing.
  • Vice versa, higher blood pressure is linked with lesser perceived stress and higher quality of life [28-30]. On the basis of the observation of an inverse relationship of blood pressure and wellbeing, Dworkin formulated the hypothesis of “learned hypertension”, i.e. blood pressure is upregulated based on the sense of an improved wellbeing [31].
  • Only more recently was aldosterone’s role in emotional regulation recognized [35,48,51-53] and its pro inflammatory role further supported [54].
  • An alternative approach to reduce aldosterone plasma concentrations and at the same time increase blood pressure was identified [203]: Glycyrrhizin, from the licorice plant, glycyrrhiza glabra, and its active metabolite glycyrrhetinic acid (enoxolone) acts as an inhibitor of the 11betaHSD2.
  • Glycyrrhizin demonstrated an effect on metabolic parameters: in an animal model (doxorubine induced cardiomyopathy) triglyceride levels decrease, whereas HDL levels increase [239], in addition to an increase in blood pressure.
  1. The following lines are too long that they have lost their meanings;
  • ‘Blood pressure has only recently been a focus of interest as predictors of depression development, i.e. as a vulnerability marker: Low blood pressure is associated with an increased risk to develop depression [33] and in particular for atypical depression, but appears to be protective against melancholic depression [34]’.
  • This is supported by data comparing outpatients with hospitalized patients with depression, the latter plausibly with more refractory forms, which show a reduced blood pressure in comparison to outpatients [37] despite the fact that waist-hip ratio and lipid levels were also higher in this population [37].
  • One potential explanation is that lower blood pressure is related to lower peripheral mineralocorticoid receptor (MR) sensitivity: we observed evidence for that in our study [35], as the correlation curve between aldosterone and blood pressure was shifted to the right in non-responders to standard treatment, i.e. a higher aldosterone concentration is required to maintain the same blood pressure.
  • A direct CNS effect may not be relevant for the action of glycyrrhizin/enoxolone in the proposed dose range, but should be briefly reviewed here for completeness: The administration of glycyrrhizin intracerebroventricularly in rats leads to an activation of the hypothalamic paraventricular nucleus, which goes along with an increase in blood pressure and renal sympathetic activity [80] by direct sensitization of the PVN neurons to cortisol/corticosterone.
  • This association between the central element of the anti-inflammatory reflex and its association with the area postrema leads to an additional important anatomical location relevant for the anti-inflammatory effect of enoxolone: TLR4 is expressed in astrocytes of the area postrema and NTS, the latter being the main target for aldosterone! Moreover, The exclamatory sign at the end of line also need justification.
  • The anti-inflammatory effect of glycyrrhizin is another im portant element of its efficacy: In a randomized placebo controlled trial with adjunct treat ment of glycyrrhizin to an SSRI an overall statistically significant benefit of glycyrrhizin vs. placebo was observed, but that effect was mainly restricted to subjects with a higher level of C-reactive protein as a sign of systemic inflammation [250].
  1. Page 11, first paragraph, first line, ‘This inhibition mediates a reduction of aldosterone release (see below for details)’. Below where? reference for figure is needed?
  2. Whole manuscript needs a thorough formatting as many lines are ended with coma, exclamatory sign or no full stop.
  3. The web link at the end of following line need justification ‘We are currently researching if enoxolone leads to changes in brain morphology in patients with depression and for which patients this may be related to treatment outcome (https://www.clinicaltrials.gov/study/NCT05570110).
  4. The font size and style for reference is different from the rest of the manuscript.
  5. The last number for reference citation in the manuscript is ‘254’ and in the reference list given at the end of the manuscript there are ‘255’ reference.
Comments on the Quality of English Language

The whole manuscript need to be checked for typos and grammatical mistakes.

Author Response

The article entitled “Discovery of personalized treatment for immune-metabolic de pression –focus on 11beta hydroxysteroid dehydrogenase type 2 (11betaHSD2) and toll like receptor 4 (TLR4) inhibition with enoxolone” written by Harald Murck. The overall manuscript has generated some useful information. However, to improve the quality of manuscript and to help readers following are the comments need to be addressed.

  1. Abstract part needs to be modified. It needs to be short, clear and to the point avoiding the extra details in section.

               HM: abstract shortened and clarified

  1. Check the journal rules for the number of keywords in the review article.

               HM: number of keywords reduced

  1. Abbreviated words such as HPA, STAR-D, HRV and many others needs to be fully pronounced at least for one time.

               HM: Abbreviations clarified.

  1. Introduction part also need modification and linking one paragraph to other including clear objectivities of the study need to be focused. The following article will help in this regard. The author needs to corelate and include these. Actas Esp Psiquiatr 2024;52(1):28-36, ISSN:1578-2735, doi: 10.62641/aep.v52i5.1748, doi: 10.34133/research.0537.

HM: A new section on the importance of the kynurenine pathway has been added to the manuscript, which covers both of the references and their topics.

  1. The following lines have some sentence or grammatical error leading to confusion in reader’s mind.

‘The fact that patients with depression have rather a low vs. high blood pressure is well established [21,22]’ is confusing.

               HM: corrected

Vice versa, higher blood pressure is linked with lesser perceived stress and higher quality of life [28-30]. On the basis of the observation of an inverse relationship of blood pressure and wellbeing, Dworkin formulated the hypothesis of “learned hypertension”, i.e. blood pressure is upregulated based on the sense of an improved wellbeing [31].

               HM: corrected

Only more recently was aldosterone’s role in emotional regulation recognized [35,48,51-53] and its pro inflammatory role further supported [54].

               HM: clarified.

An alternative approach to reduce aldosterone plasma concentrations and at the same time increase blood pressure was identified [203]: Glycyrrhizin, from the licorice plant, glycyrrhiza glabra, and its active metabolite glycyrrhetinic acid (enoxolone) acts as an inhibitor of the 11betaHSD2.

               HM: modified.

Glycyrrhizin demonstrated an effect on metabolic parameters: in an animal model (doxorubine induced cardiomyopathy) triglyceride levels decrease, whereas HDL levels increase [239], in addition to an increase in blood pressure.

  1. The following lines are too long that they have lost their meanings;

‘Blood pressure has only recently been a focus of interest as predictors of depression development, i.e. as a vulnerability marker: Low blood pressure is associated with an increased risk to develop depression [33] and in particular for atypical depression, but appears to be protective against melancholic depression [34]’.

This is supported by data comparing outpatients with hospitalized patients with depression, the latter plausibly with more refractory forms, which show a reduced blood pressure in comparison to outpatients [37] despite the fact that waist-hip ratio and lipid levels were also higher in this population [37].

               HM: modified.

One potential explanation is that lower blood pressure is related to lower peripheral mineralocorticoid receptor (MR) sensitivity: we observed evidence for that in our study [35], as the correlation curve between aldosterone and blood pressure was shifted to the right in nonresponders to standard treatment, i.e. a higher aldosterone concentration is required to maintain the same blood pressure.

               HM: modified

A direct CNS effect may not be relevant for the action of glycyrrhizin/enoxolone in the proposed dose range, but should be briefly reviewed here for completeness: The administration of glycyrrhizin intracerebroventricularly in rats leads to an activation of the hypothalamic paraventricular nucleus, which goes along with an increase in blood pressure and renal sympathetic activity [80] by direct sensitization of the PVN neurons to cortisol/corticosterone.

               HM: modified

This association between the central element of the anti-inflammatory reflex and its association with the area postrema leads to an additional important anatomical location relevant for the anti-inflammatory effect of enoxolone: TLR4 is expressed in astrocytes of the area postrema and NTS, the latter being the main target for aldosterone! Moreover, The exclamatory sign at the end of line also need justification.

HM: modified.

The anti-inflammatory effect of glycyrrhizin is another important element of its efficacy: In a randomized placebo controlled trial with adjunct treatment of glycyrrhizin to an SSRI an overall statistically significant benefit of glycyrrhizin vs. placebo was observed, but that effect was mainly restricted to subjects with a higher level of C-reactive protein as a sign of systemic inflammation [250].

               HM: modified

  1. Page 11, first paragraph, first line, ‘This inhibition mediates a reduction of aldosterone release (see below for details)’. Below where? reference for figure is needed?

               HM: text reorganized

  1. Whole manuscript needs a thorough formatting as many lines are ended with coma, exclamatory sign or no full stop.

               HM: done

  1. The web link at the end of following line need justification ‘We are currently researching if enoxolone leads to changes in brain morphology in patients with depression and for which patients this may be related to treatment outcome (https://www.clinicaltrials.gov/study/NCT05570110).

               HM: clarified

  1. The font size and style for reference is different from the rest of the manuscript.

               HM: corrected

  1. The last number for reference citation in the manuscript is ‘254’ and in the reference list given at the end of the manuscript there are ‘255’ reference.

HM: modified. This was due to the fact that the legend of one figure was moved into the flow of the manuscript.

Comments on the Quality of English Language

The whole manuscript need to be checked for typos and grammatical mistakes

               HM: careful review done.

Reviewer 2 Report

Comments and Suggestions for Authors

In this review, the authors describe that approximately one-third of patients with depression exhibit features such as metabolic disturbances, inflammatory markers, and alterations in brain morphology. They also summarize the biological roles of glycyrrhizin and its metabolite enoxolone. Enoxolone is reported to inhibit 11betaHSD2 and TLR4, thereby modulating innate immunity. Evidence from clinical trials further supports these findings.

I would like to offer the following suggestions for the authors:

  1. Please consider adding a table to clearly summarize the role of enoxolone in inhibiting 11betaHSD2 and TLR4.
  2. Please discuss other reported targets of enoxolone, as these may represent potential off-target effects relevant to its clinical application.

Author Response

I would like to offer the following suggestions for the authors:

Please consider adding a table to clearly summarize the role of enoxolone in inhibiting 11betaHSD2 and TLR4.

               HM: we provided two diagrams (Fig 2a and 2b), which hopefully serves to clarify the mechanism.

Please discuss other reported targets of enoxolone, as these may represent potential off-target effects relevant to its clinical application.

HM: we discussed other potential mechanisms, which may contribute to the biological effect of enoxolone, in the “limitations’ section.

Round 2

Reviewer 1 Report

Comments and Suggestions for Authors

The authors have addressed most of my comments and the article is now much modified and can be accepted